# Structure and conformational states of the bovine mitochondrial ATP synthase by cryo-EM

Anna Zhou[1,2†], Alexis Rohou[3†], Daniel G Schep[1,2], John V Bason[4], Martin G Montgomery[4], John E Walker[4*], Nikolaus Grigorieff[3*], John L Rubinstein[1,2,5*]

[1]The Hospital for Sick Children Research Institute, Toronto, Canada; [2] Department of Medical Biophysics , The University of Toronto, Ontario, Canada; [3]Janelia Research Campus, Howard Hughes Medical Institute, Ashburn, United States; [4]MRC Mitochondrial Biology Unit, Cambridge, United Kingdom; [5] Department of Biochemistry , The University of Toronto, Ontario, Canada

**Abstract** Adenosine triphosphate (ATP), the chemical energy currency of biology, is synthesized in eukaryotic cells primarily by the mitochondrial ATP synthase. ATP synthases operate by a rotary catalytic mechanism where proton translocation through the membrane-inserted $F_O$ region is coupled to ATP synthesis in the catalytic $F_1$ region via rotation of a central rotor subcomplex. We report here single particle electron cryomicroscopy (cryo-EM) analysis of the bovine mitochondrial ATP synthase. Combining cryo-EM data with bioinformatic analysis allowed us to determine the fold of the a subunit, suggesting a proton translocation path through the $F_O$ region that involves both the a and b subunits. 3D classification of images revealed seven distinct states of the enzyme that show different modes of bending and twisting in the intact ATP synthase. Rotational fluctuations of the $c_8$-ring within the $F_O$ region support a Brownian ratchet mechanism for proton-translocation-driven rotation in ATP synthases.

*For correspondence: walker@ mrc-mbu.cam.ac.uk (JEW); niko@ grigorieff.org (NG); john. rubinstein@utoronto.ca (JLR)

[†]These authors contributed equally to this work

Competing interests: The authors declare that no competing interests exist.

## Introduction

In the mitochondria of eukaryotes, adenosine triphosphate (ATP) is produced by the ATP synthase, a ~600 kDa membrane protein complex composed of a soluble catalytic $F_1$ region and a membrane-inserted $F_O$ region. The ATP synthase is found in the inner membranes of mitochondria, with the $F_1$ region in the mitochondrial matrix and the $F_O$ region accessible from the inter-membrane space between the mitochondrial outer and inner membranes. In the mammalian enzyme, the subunit composition is $\alpha_3\beta_3\gamma\delta\epsilon$ for the $F_1$ region with subunits a, e, f, g, A6L, DAPIT, a 6.8 kDa proteolipid, two membrane-inserted $\alpha$-helices of subunit b, and the $c_8$-ring forming the $F_O$ region (*Walker, 2013*). The rotor subcomplex consists of subunits $\gamma$, $\delta$, $\epsilon$, and the $c_8$-ring. In addition to the rotor, the $F_1$ and $F_O$ regions are linked by a peripheral stalk composed of subunits OSCP, d, $F_6$, and the hydrophilic portion of subunit b. Approximately 85% of the structure of the complex is known at high resolution from X-ray crystallography of constituent proteins, which have been assembled into a mosaic structure within the constraints of a cryo-EM map at 18 Å resolution (*Walker, 2013*; *Baker et al., 2012*).

The proton motive force, established by the electron transport chain during cellular respiration, drives protons across the $F_O$ region through the interface between the a subunit and the $c_8$-ring, inducing rotation of the rotor (*Boyer, 1997*; *Walker, 1998*). While the mechanism by which ATP synthesis and hydrolysis are coupled to rotation of the $\gamma$ subunit is now understood well (*Walker, 2013*), it is still unresolved how rotation of the central rotor is coupled to proton translocation through the

**eLife digest** A molecule called adenosine triphosphate (ATP) is the energy currency in cells. Most of the ATP used by cells is made by the membrane-embedded enzyme ATP synthase. This enzyme is found in membranes inside specialized compartments known as mitochondria. ATP synthase is made up of many protein subunits that work together as a molecular machine. Hydrogen ions flow across the membrane through the ATP synthase, turning a rotor structure within the enzyme, which leads to the production of ATP.

It is not known how the transport of hydrogen ions causes rotation of the rotor. Some researchers have proposed that the enzyme works as a ratchet that is driven by the random Brownian motion of the rotor. That is, the rotational position of the rotor fluctuates randomly, but a ratchet mechanism ensures that there is a net rotation in one direction. However, there is currently little experimental evidence to back up this theory, which is known as the Brownian ratchet model.

Zhou, Rohou et al. used a technique called electron cryomicroscopy (or cryo-EM) to study ATP synthase from cows. The cryo-EM data made it possible to use computer software to construct a three-dimensional model of the enzyme that is more detailed than previous attempts. Zhou, Rohou et al. show that the structure of ATP synthase is flexible, with the different protein subunits bending, flexing, and rotating relative to each other. This variability in the position of the rotor is consistent with the Brownian ratchet model.

Together, these findings reveal important new details about the structure of ATP synthase and provide some of the first experimental evidence for the Brownian ratchet model. The new three-dimensional structure of ATP synthase will open the door to testing hypotheses of how the ATP synthase works.

$F_O$ region. The most popular model suggests that proton translocation occurs through two offset half channels near the a subunit/c subunit interface (*Junge et al., 1997*, *Junge, 2005*). In this model, one half channel allows protons to move half-way across the lipid bilayer in order to protonate the conserved Glu58 residue of one of the c subunits. The other half channel allows deprotonation of the adjacent c subunit (*Lau and Rubinstein, 2012*), setting up the necessary condition for a net rotation of the entire c-ring. Rotation does not occur directly from the protonating half channel to the deprotonating half channel, but in the opposite direction so that the protonated, and therefore uncharged, Glu residues traverse through the lipid environment before reaching the deprotonating half channel. The deprotonated Glu residue prevents the ring from rotating in the opposite direction, which would place the charged residue in the hydrophobic environment of the lipid bilayer. Rotation of the c-ring occurs due to Brownian motion, making the enzyme a Brownian ratchet.

A recent cryo-EM map of the *Polytomella* sp. ATP synthase dimer showed two long and tilted α-helices from the a subunit in contact with the $c_{10}$-ring of that species (*Allegretti et al., 2015*). This arrangement of α-helices from the a and c subunits was also seen in the *Saccharomyces cerevisiae* V-type ATPase (*Zhao et al., 2015a*). Cryo-EM of the *S. cerevisiae* V-ATPase demonstrated that images of rotary ATPases could be separated by 3D classification to reveal conformations of the complex that exist simultaneously in solution. In the work described here, we obtained and analyzed cryo-EM images of the bovine mitochondrial ATP synthase. 3D classification of the images resulted in seven distinct maps of the enzyme, each showing the complex in a different conformation. By averaging the density for the proton-translocating a subunit from the seven maps, we generated a map segment that shows α-helices clearly. Analysis of evolutionary covariance in the sequence of the a subunit (*Göbel et al., 1994*) allowed the entire a subunit polypeptide to be traced through the density map. The resulting atomic model for the a subunit was fitted into the maps for the different rotational states, suggesting a path for protons through the enzyme and supporting the Brownian ratchet mechanism for the generation of rotation (*Junge et al., 1997*; *2005*), and thereby ATP synthesis, in ATP synthases.

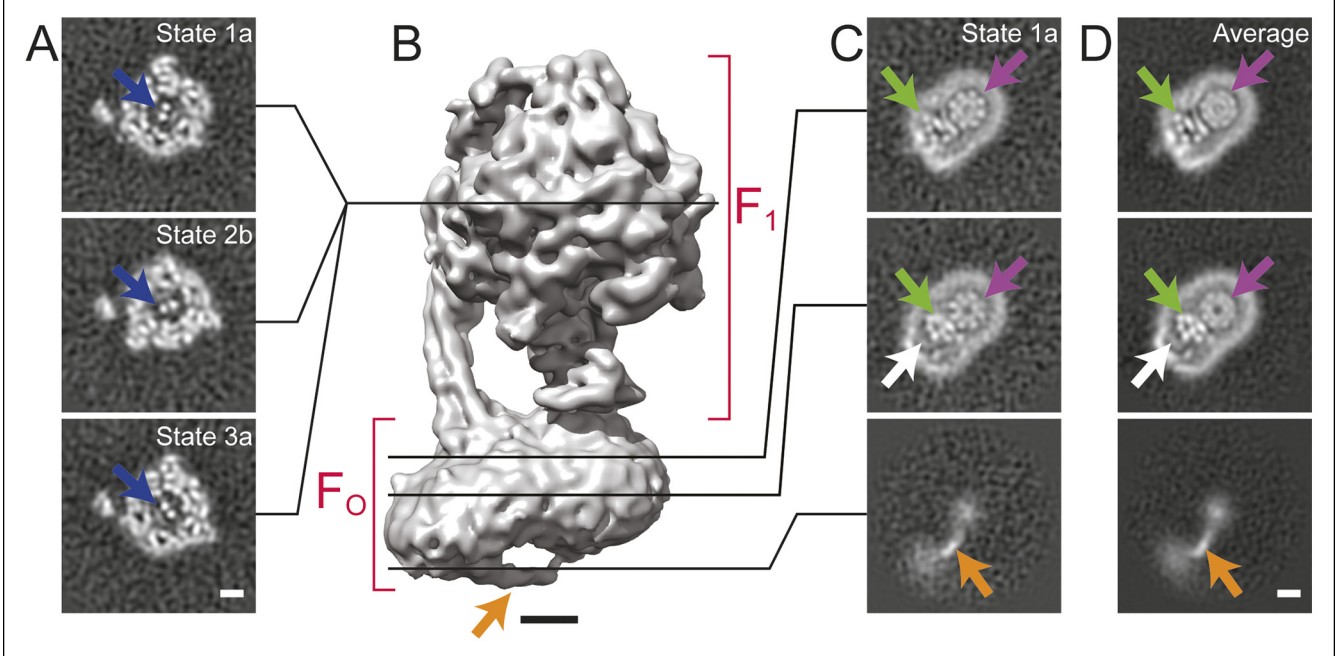

**Figure 1.** Cross-sections through maps. (**A**) Cross-sections through the $F_1$ regions of the different maps show that states 1, 2, and 3 are related by ~120° rotations of the γ subunit within the $\alpha_3\beta_3$ hexamer (blue arrows). (**B**) Surface rendering of a map (State 1a) shows the bent $F_O$ region with a tubular feature that extends from the rotor-distal portion to the $c_8$-ring (orange arrow). (**C**) Cross-sections through the $F_O$ region show α-helices from the a, b and A6L subunits (green arrows), a low density region in the rotor-distal portion (white arrow), the tubular extension from the rotor-distal portion to the rotor (orange arrows), and the c-ring (purple arrows). (**D**) Averaging the $F_O$ regions from the seven different maps shows all of the features mentioned above with an improved signal-to-noise ratio for some features. Scale bars, 25 Å.

The following figure supplements are available for figure 1:

**Figure supplement 1.** Electron microscopy and model construction.

**Figure supplement 2.** The seven observed states of the bovine mitochondrial ATP synthase.

**Figure supplement 3.** Fourier shell correlation curves for the seven maps.

## Results

Specimens of ATP synthase were isolated from bovine heart mitochondria and prepared for cryo-EM as described previously (*Baker et al., 2012*; *Runswick et al., 2013*) (*Figure 1—figure supplement 1*). Initial 3D classification produced three classes, each of which appear to show a ~120° rotation of the central rotor within the $F_1$ region of the complex (*Figure 1A*, blue arrows), similar to what was seen previously with the *S. cerevisiae* V-ATPase (*Zhao, et al., 2015a*). Further classification of these three rotational states was able to separate state 1 into two sub-states, subsequently referred to as states 1a and 1b. State 2 could be divided into states 2a, 2b, and 2c, while state 3 could be separated into states 3a and 3b. Each of these 3D classes shows a different conformation of the enzyme (*Figure 1—figure supplement 2* and *Video 1*). While the rotational states of the yeast V-ATPase were found to be populated unequally after 3D classification, bovine ATP synthase classes corresponding to different positions of the rotor had approximately equal populations. State 1 contained 43,039 particle images divided almost equally over its two sub-states, state 2 contained 48,053 particle images divided almost equally over its three sub-states, and state 3 contained 46,257 particle images divided almost equally over its two sub-states. The resolutions of the seven classes were between 6.4 and 7.4 Å (*Figure 1—figure supplement 3*).

There is a distinct bend in the $F_O$ region of the complex between the portion that is proximal to the $c_8$-ring and the portion that is distal to the $c_8$-ring (*Figure 1B* and *Figure 1—figure supplement 2*). This bent structure was seen previously in a lower-resolution cryo-EM map of the bovine

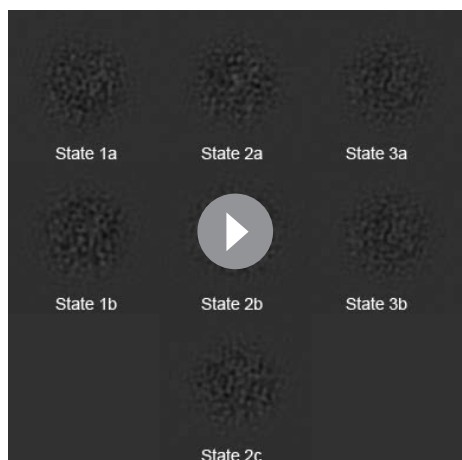

**Video 1.** Slices through the seven cryo-EM maps. Cross-sections are shown moving from the $F_1$ region towards the $F_O$ region for states 1a, 1b, 2a, 2b, 2c, 3a, and 3b.

mitochondrial ATP synthase (*Baker et al., 2010*). It is consistent with electron tomograms of ATP synthases in mitochondrial membranes (*Strauss, 2008*; *Davies et al., 2011*) and was also observed recently by electron tomography of membrane-reconstituted 2D crystals of the bovine enzyme (*Jiko et al., 2015*). The e and g subunits are expected to reside in the portion of the $F_O$ region distal to the $c_8$-ring because cryo-EM maps of the mitochondrial ATP synthase from *S. cerevisiae*, where subunits e and g were removed by detergent, lacked this bent portion (*Baker et al., 2010*; *Lau et al., 2008*). The f subunit is also thought to be associated with the e and g subunits (*Belogrudov et al., 1996*). DAPIT and the 6.8 kDa proteolipid are not expected to be present in this preparation because the necessary lipids for maintaining their association were not added during purification (*Chen et al., 2007*; *Carroll, 2009*). While a detergent micelle can be seen around the entire $F_O$ region, the portion of $F_O$ distal to the $c_8$-ring also contains a feature with unusually low density (*Figure 1C*, white arrows). The content of this low-density region is uncertain. Low density in cryo-EM maps is often due to partial occupancy, flexibility, or disorder of a protein subunit. However, the low-density feature here is bounded on one side by unusually sharp density from the detergent micelle, suggesting more order than in the rest of the micelle, and on the other side by the a subunit, which also appears well ordered. Therefore, the low density region could be due to bound material with low density, possibly lipid, that remains after purification of the enzyme.

## A novel feature in the $F_O$ region

The $F_O$ regions of all seven maps also revealed a remarkable feature not resolved previously in cryo-EM maps of ATP synthases (*Baker et al., 2012*; *Allegretti et al., 2015*; *Lau et al., 2008*; *Rubinstein et al., 2003*). The feature appears to consist of an elongated membrane-embedded density, possibly an α-helix, that extends from the rotor-distal portion of $F_O$ to the $c_8$-ring. The orientation of this density would cause it to pass through the inter-membrane space of the mitochondrion (*Figure 1B and C*, orange arrow). While not identified in the previous cryo-EM map of the enzyme at 18 Å resolution, the structure is consistent with a poorly-resolved ridge along the surface of the $F_O$ region seen in the earlier map (*Baker et al., 2012*). Because it extends from the bent end of the $F_O$ region, this feature may correspond to the soluble part of the e subunit. Indeed, a similar structure was observed in single particle EM of negatively stained ATP synthase dimers from bovine heart mitochondria, and was proposed to be interacting e subunits (*Minauro-Sanmiguel et al., 2005*). However, in the present structure the feature is not positioned to interact between dimers of the enzyme and its role in the complex remains unclear.

## Subunit a, b and A6L in the $F_O$ region

In order to improve the signal-to-noise ratio for the $F_O$ region of the complex, the membrane regions from the seven different maps were aligned and averaged. Averaging maps increases the signal-to-noise ratio where the structures are similar, but blurs regions where the maps differ. In principle, this method could also be applied to other map regions of the ATP synthase or other heterogeneous protein complexes by applying an appropriate transform before averaging. Averaging the $F_O$ region provides a clear view of the portion of the $F_O$ region adjacent to the rotor, allowing the trans-membrane α-helices from the a, b, and A6L subunits to be identified reliably (*Figure 1C and D*, green arrows, and *Figure 2*). The c-ring has a lower density in the averaged membrane region than in the original maps, suggesting that its position differs between maps (*Figure 1C and D*,

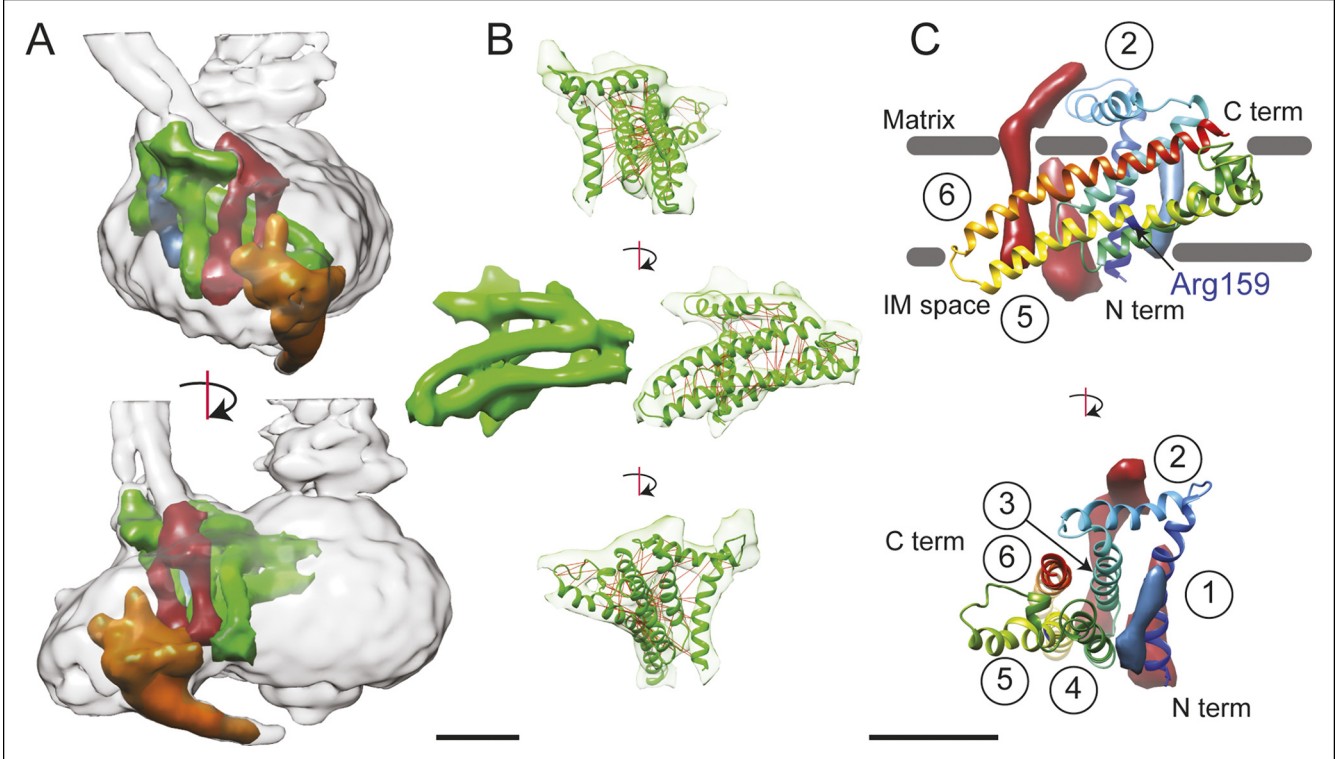

**Figure 2.** 3D structure of the $F_O$ region. (**A**) In the $F_O$ region of the complex, density was segmented for the a subunit (green), the b subunit (red), the A6L subunit (blue), and the structure thought to arise from subunits e and g (orange). (**B**) The a subunit sequence could be placed unambiguously into the cryo-EM density (green) by including constraints for residues predicted to be near to each other due to evolutionary covariance (red lines). (**C**) The a subunit (coloured with a gradient from blue to red to denote directionality from the N to C terminus) possesses six α-helices, numbered 1–6. Trans-membrane α-helices from subunits b and A6L are shown as volumes (red and blue, respectively). Five of the α-helices of subunit a are membrane-inserted while helix #2 runs along the matrix surface of the $F_O$ region. The N terminus of the a subunit is on the inter-membrane space side of the subunit while the C terminus is on the matrix side. The highly conserved residue Arg159 is on the elongated and highly tilted α-helix #5. Scale bar, 25 Å.

The following figure supplement is available for figure 2:

**Figure supplement 1.** Analysis of evolutionary covariance of residues.

purple arrows). The averaged density for the $F_O$ region revealed the a subunit to have five membrane-inserted α-helices and an additional α-helix along the plane of the membrane surface (*Figure 2*). Three additional trans-membrane α-helices are also apparent, presumably two from the b subunit (*Walker et al., 1987*) and one from the A6L subunit (*Fearnley and Walker, 1986*). The mammalian mitochondrial a subunit possesses the two highly tilted α-helices in contact with the c-ring that were seen previously for the *Polytomella* sp. F-type ATP synthase (*Allegretti et al., 2015*) and *S. cerevisiae* V-ATPase (*Zhao et al., 2015a*) (*Figure 2A*).

A model for the a subunit was built into the cryo-EM density map using constraints from analysis of evolutionary covariance in sequences of the a subunit from different species. Analysis of covariance in evolutionarily related protein sequences can identify pairs of residues in a protein structure that are likely to interact physically with each other (*Göbel et al., 1994*; *Cronet et al., 1993*; *Hopf et al., 2012*; *Ovchinnikov et al., 2014*). Spatial constraints from covariance analysis were sufficient not only to identify tentatively trans-membrane α-helices of the a subunit that are adjacent to each other, but also suggest which face each α-helix presents to the other α-helices (*Figure 2B* and *Video 2*, red lines). The constraints show patterns of interaction consistent with the predicted α-helical structure of the a subunit (*Figure 2—figure supplement 1A*), as well as interactions between the a subunit and the outer α-helix of a c subunit in the $c_8$-ring (*Figure 2—figure supplement 1B*). As a result, we were able to trace the path of the a subunit polypeptide through the cryo-EM density

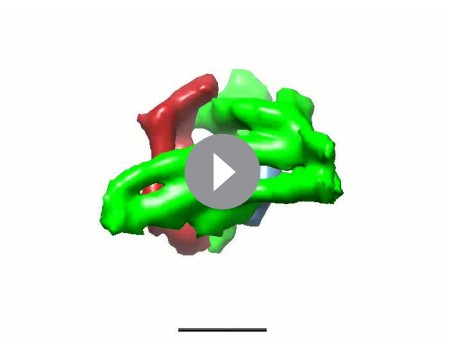

**Video 2.** Fold of the a subunit. The density corresponding to the a subunit (green), membrane-inserted portion of the b subunit (red), and A6L subunit (blue) are shown, in addition to a ribbon diagram for the a subunit (green) and the top 90 constraints from analysis of covarying residues in the a subunit sequence (red lines). 6% of the constraints could not be satisfied, which is consistent with the false positive rate from known structures (*Marks et al., 2011*). Scale bar, 25 Å.

map. The fit of the α-helices in the a subunit density was improved by molecular dynamics flexible fitting (*MDFF*) (*Trabuco et al., 2008*) and the 34 residue long connecting loop from residues 115 to 148 was built with *Rosetta* (*Rohl et al., 2004*) (*Figure 2B* and *Video 2*). This connecting loop was built to be physically reasonable, but because its structure is not derived from experimental data it is not included in the discussion below. The final model places the α carbons of the residues in the co-varying pairs within 15 Å of each other in 94% of the top 90 identified pairs, with an average $C_\alpha$ to $C_\alpha$ distance of 10.3 Å. The 6% of constraints that are violated by the model are consistent with the false positive rate observed when testing covariance analysis approaches with proteins of known structure (*Marks et al., 2011*).

## Description of the a subunit structure

The mammalian a subunit appears to consist of six α-helices, with five α-helices that penetrate into the membrane (*Figure 2C*). The N terminus of the subunit is in the inter-membrane space of the mitochondrion. The first α-helix extends vertically across the $F_O$ region distal to the contact of the a subunit and $c_8$-ring. The two trans-membrane α-helices of the b subunit are packed against one surface of helix #1 while the single trans-membrane α-helix from the A6L subunit is packed against its opposite surface. The second density region, interpreted as an α-helix of the a subunit, is not membrane-inserted and extends along the matrix surface of the $F_O$ region connecting the membrane-inserted α-helix #1 with a membrane-inserted helical-hairpin composed of α-helices #3 and #4. This hairpin of the third and fourth α-helices does not appear to cross the $F_O$ region fully, as seen previously in the *Polytomella* sp. ATP synthase (*Allegretti et al., 2015*). The final two trans-membrane helices are the two highly tilted α-helices seen previously with the *Polytomella* sp. ATP synthase and *S. cerevisiae* V-ATPase, with the C terminus of the a subunit on the matrix side of the $F_O$ region. Within this structure, Arg159, which is essential and completely conserved, is found near the middle of the long tilted α-helix #5 nearer the inter-membrane space side of the $F_O$ region, different from its predicted position in the *Polytomella* sp. enzyme (*Allegretti et al., 2015*).

## Docking of atomic models into the cryo-EM maps

To analyze the different enzyme conformations detected by 3D classification, the maps were segmented and available crystal structures for the $F_1$:$IF_1$ complex (*Gledhill et al., 2007*), $F_1$ peripheral stalk complex (*Rees et al., 2009*), peripheral stalk alone (*Dickson et al., 2006*), and $F_1$-$c_8$ complex (*Watt et al., 2010*) were combined into each of the maps by *MDFF* (*Trabuco et al., 2008*). Residues for the b subunit were extended from the N terminus of the b subunit crystal structure into the membrane region based on trans-membrane α-helix prediction. While *MDFF* with maps in this resolution range cannot be used to determine the locations or conformations of amino acid side chains, loops, or random-coil segments of models, it can show the positioning of α-helices in the structures. *Figures 3A and B* compare the fitting for state 1a (*Figure 3A*) and state 1b (*Figure 3B*), illustrating the accuracy with which α-helical segments could be resolved in the maps of different sub-states. The atomic model alone for state 1a is shown in *Figure 3C*, with the $c_8$-ring removed for clarity in *Figure 3D*. Transitions between the different states were illustrated by linear interpolation (*Video 3*). As seen previously for the *S. cerevisiae* V-ATPase, almost all of the subunits in the enzyme undergo conformational changes on transition between states (*Zhao, et al., 2015a*). Because there were two sub-states identified for states 1 and 3 there is only a single sub-state to sub-state transition for

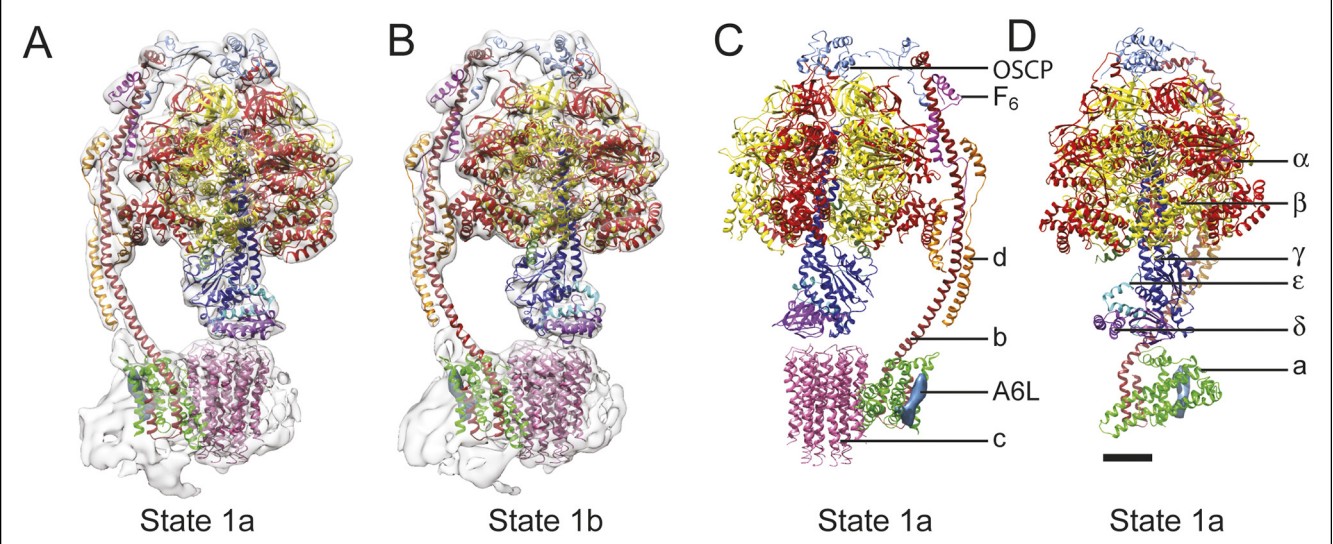

**Figure 3.** Docking of atomic models into the cryo-EM maps. Fitting of all available atomic models into the density map is shown for state 1a (**A**) and state 1b (**B**). State 1a is also shown in a different orientation and without the density map (**C**) and with the $c_8$-ring removed for clarity (**D**). The apparent gap between the $c_8$-ring and γ and δ subunits is filled with amino acid side chains and is the same as was seen in the crystal structure of the $F_1$-$c_8$ complex (**Watt et al., 2010**). Scale bar, 25 Å.

The following figure supplement is available for figure 3:

**Figure supplement 1.** Differences between sub-states.

these two states. In comparison, three different sub-states were identified for state 2 and consequently there are three sub-state to sub-state transitions that are possible for this state. All of the sub-state to sub-state transitions include a slight rotation of the $c_8$-ring against the a subunit. It is possible that this movement is due to partial disruption of the subunit a/$c_8$-ring interface. However, the structural differences within the $F_O$ regions of different classes are significantly smaller than the structural differences seen elsewhere in the enzyme, suggesting that these changes do not originate from disruption within the membrane region of the complex and instead reflect flexibility in the enzyme.

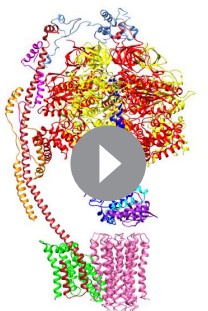

**Video 3.** Conformation changes during the rotary cycle. Linear interpolation is shown between one of each of the main states identified by 3D classification (state 1a, 2a, and 3a) showing the large conformation changes that occur during rotation. Scale bar, 25 Å. [*please view movie as loop*]

The largest change between the sub-states of each state can be approximated by rotations of the $\alpha_3\beta_3$ hexamer relative to the rest of the complex by angles ranging from 10 to 16°. A comparison of the maps for the different sub-states and the axes of these rotations are shown in *Figure 3—figure supplement 1*. The resulting conformational changes can be seen most clearly in *Videos 4* and *5*. The state 1a to 1b transition reveals a bending of the peripheral stalk towards the top of the $F_1$ region near the OSCP and $F_6$ subunits (*Videos 4* and *5*, panel A). In comparison, the state 3a to 3b transition reveals bending of the peripheral stalk near where the b subunit enters the membrane (*Videos 4* and *5*, panel D). Transitions between the three sub-states of state 2 show both motions: the transition between 2a and 2b shows mostly bending of the peripheral stalk near OSCP and $F_6$ subunits, while the 2b and 2c transition shows mostly bending near the

membrane-inserted portion of the peripheral stalk (*Videos 4* and *5*, panels B and C, respectively). The transition from state 2a to 2c shows a combined bending at both of these positions. It is most likely that the different modes of bending exist in all of the states and further classification of larger datasets would be expected to reveal these complex motions. The different sub-states do not appear to have a specific sequence or represent specific intermediates in the catalytic rotation sequence. Instead, the differences in conformation between sub-states when taken together illustrate the flexibility of the enzyme, a property that has been linked to its rapid rate of enzymatic activity (*Zhao, et al., 2015a*; *Zhou et al., 2014*). The functional significance of the sub-states may also be determined by the orientation of the $c_8$-ring with respect to the a subunit, as discussed below.

## Discussion

Predicting the path of protons through membrane protein complexes has proven difficult, even in cases where high-resolution atomic models including bound water molecules are available from X-ray crystallography (*Hosler et al., 2006*). Nonetheless, features in the structure of the bovine mitochondrial $F_O$ region suggest a possible path for proton translocation similar to a model put forward based on the structure of the *Polytomella* sp. ATP synthase (*Allegretti et al., 2015*). The arrangement of α-helices in the $F_O$ region is remarkably similar to the arrangement of α-helices in the $V_O$ region of the yeast V-ATPase (*Zhao, et al., 2015a*), even though the V-ATPase a subunit has eight α-helices and little detectable sequence similarity with the F-type ATP synthase a subunit. The conserved general architecture of the membrane-inserted regions in F-type ATP synthases and V-type ATPases suggests that the observed arrangement of α-helices is functionally important and likely involved in proton translocation (*Figure 4A and B*). The matrix half channel of the ATP synthase is likely to be formed by the cavity between the $c_8$-ring and the matrix ends of tilted α-helices #5 and #6 of the a subunit. The lumenal half channel in the V-ATPase is probably formed entirely from α-helices from the a subunit, whereas the corresponding inter-membrane space half channel in the ATP synthase is likely composed of the inter-membrane space ends of α-helices #5 and #6 and one or both of the two trans-membrane α-helices of the b subunit. Defining the exact placement of half channels will likely require higher-resolution maps from cryo-EM or X-ray crystallography that reveal amino acid side chain density and bound water molecules.

In addition to bending and twisting of the peripheral stalk and central rotor of the enzyme, the differences between the sub-states of each state show variability in the rotational position of the $c_8$-ring in relation to the a subunit (*Figure 4C*

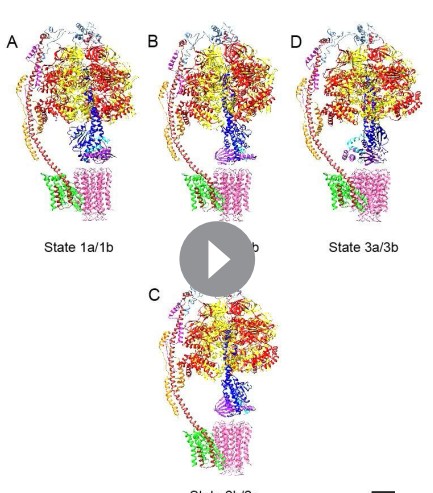

**Video 4.** Conformational differences between the different ATP synthase maps. Differences between conformations detected by 3D classification are illustrated by linear interpolation between state 1a and 1b, showing bending of the peripheral stalk near the OSCP and $F_6$ subunits (**A**), state 2a and 2b showing a similar bending of the peripheral stalk near the OSCP and $F_6$ subunits (**B**), state 2b and 2c showing bending of the peripheral stalk near the membrane region (**C**), and state 3a and 3b showing a similar bending of the peripheral stalk near the membrane region (**D**). Scale bar, 25 Å. [*please view movie as loop*]

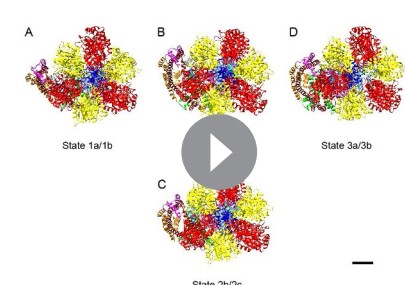

**Video 5.** Conformational differences between the different ATP synthase maps. The same interpolations as shown in *Video 4*, except viewed from the $F_1$ region towards the $F_O$ region. Scale bar, 25 Å. [*please view movie as loop*]

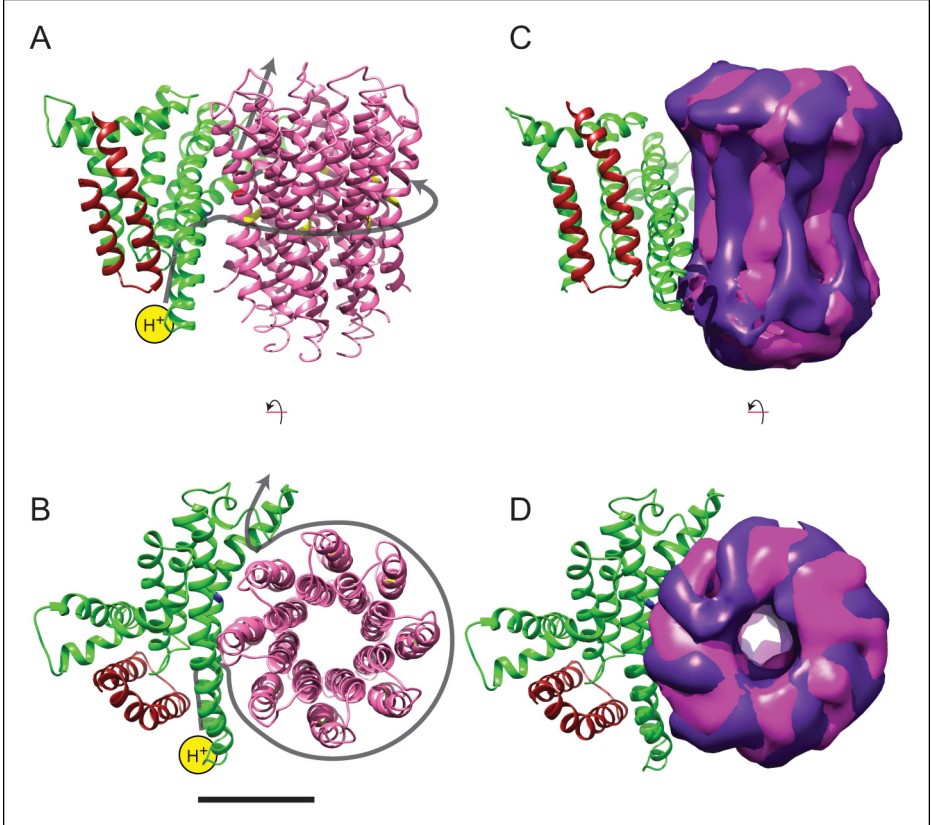

**Figure 4.** Model for proton translocation. (A and B) The a subunit (green), along with the membrane-intrinsic α-helices of the b subunit (red), form two clusters that could be the half channels needed for trans-membrane proton translocation. (C and D) The map segment corresponding to the $c_8$-ring is shown for state 2a (pink) and state 2c (purple). The difference in rotational position of the c-ring is consistent with the Brownian fluctuations predicted for the generation of a net rotation. Scale bar, 25 Å.

*and D*), even in the nucleotide-depleted conditions in which cryo-EM grids were frozen for this analysis. This lack of a rigid interaction between the $c_8$-ring and a subunit is consistent with the Brownian ratchet model of proton translocation (*Junge et al., 1997*). In the Brownian ratchet model, the rotational position of the ring fluctuates due to Brownian motion, but the ring cannot turn to place the Glu58 residue of a c subunit into the hydrophobic environment of the lipid bilayer until the Glu58 is protonated at the inter-membrane space half channel. Therefore, with this model, the different substates would correspond to energetically equivalent or nearly-equivalent conformations that occur due to Brownian motion. *Video 6* illustrates the extent of rotational oscillation predicted from the transition between states 2a and 2c. It is most likely that this oscillation occurs as each c subunit passes the interface with the a subunit, with 8/3 c subunits on average contributing to the synthesis of one ATP molecule. The rotational flexibility of the $c_8$-ring that exists even when the γ subunit is locked within the $α_3β_3$ hexamer suggests that flexing and bending of the components of the ATP synthase smooths the coupling of the 8-step rotation of the $c_8$-ring with the 3-step rotation of the $F_1$ region. This model suggests that the observed flexibility in the enzyme, which apparently complicates determination of atomic resolution structures directly from cryo-EM data, is also essential to the mechanism of ATP synthesis.

## Materials and methods

### Protein purification and electron microscopy

Bovine mitochondrial ATP synthase was purified as described previously (*Runswick et al., 2013*) and cryo-EM specimen grids were prepared as described previously, except that glycerol was removed

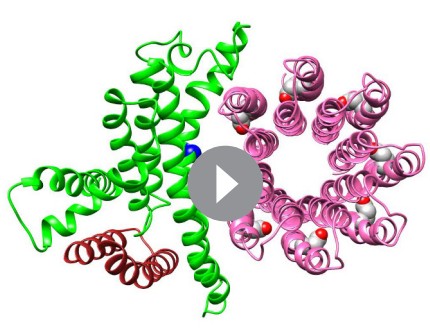

**Video 6.** Brownian ratcheting. The different rotational positions of the $c_8$-ring between the sub-states are illustrated by interpolating between states 2a and 2c. The conserved residue Arg159 is shown as a blue sphere. The movement is consistent with the Brownian ratcheting predicted during proton translocation in rotary ATPases. Glu58 residues are shown moving from the proton-locked conformation (*Pogoryelov et al., 2009*) to an open conformation (*Pogoryelov et al., 2010*; *Symersky et al., 2012*) when they are close to the conserved Arg159 residue. Scale bar, 25 Å. [*please view movie as loop*]

from specimens prior to grid freezing with a 7 kDa molecular weight cutoff Zeba Spin centrifuged desalting column (Thermo Scientific) and nano-fabricated grids with 500 nm holes were used (*Marr et al., 2014*). After optimization of grid freezing conditions, micrographs were recorded from three grids on a Titan Krios microscope (FEI) operated at 300 kV with parallel illumination of a 2.5 μm diameter area of the specimen and an electron fluency of 3 el⁻/Å²/s. A 70 μm objective aperture was employed with a nominal magnification of 18,000 × onto a K2 Summit direct detector device (Gatan Inc.) operated in super-resolution mode with a 1.64 Å physical pixel and 0.82 Å super-resolution pixel. With no specimen present, the rate of exposure of the detector was 8 el⁻/pixel/s. Exposure-fractionated movies of 20.1 s were recorded as stacks of 67 frames, so that selected specimen areas were exposed with a total of 60.3 el⁻/Å². Data collection was automated with *SerialEM* (*Mastronarde, 2005*).

## Image processing

Magnification anisotropy (*Zhao et al., 2015b*) under the conditions described above was measured previously from images of a standard cross-grating specimen with the program *mag_distortion_estimate* (*Grant and Grigorieff, 2015a*). The linear scaling parameters were 0.986 and 1.013, the azimuth of the distortion was 134.0°, and the program *mag_distortion_correct* was used to correct for this distortion in each dose-fractionated frame. The frames were then down-sampled to a pixel size of 1.64 Å by Fourier-space cropping and aligned with each other with the program *Unblur* (*Grant and Grigorieff, 2015a*). Defocus parameters were estimated from aligned sums of frames using *CTFFIND4* (*Rohou and Grigorieff, 2015*). Particle images were selected in *Relion* and subjected to 2D classification (*Scheres, 2015*; *Scheres, 2012*), yielding a set of 195,233 single particle coordinates selected from 5,825 movies. Local beam-induced motion was corrected for each particle with the program *alignparts_lmbfgs* (*Rubinstein and Brubaker, 2015*). Aligned dose-fractionated particle images were filtered and summed to optimize the signal-to-noise ratio at all spatial frequencies (*Grant and Grigorieff, 2015b*; *Rubinstein and Brubaker, 2015*; *Baker et al., 2010*), giving a set of particle images that were 256 × 256 pixels. These images were down-sampled to 128 × 128 pixels (pixel size of 3.28 Å) for determining particle orientations.

Initial single-particle alignment parameter values were obtained by 5 rounds of iterative grid search and reconstruction in *FREALIGN*'s mode 3 (*Grigorieff, 2007*), using the earlier published map of the enzyme as an initial reference (*Baker et al., 2012*). *FREALIGN*'s likelihood-based classification algorithm (*Lyumkis et al., 2013*) was then used to classify particles images into several maps, alternating between refinement of orientation parameters every 3rd or 4th iteration and class occupancy during other iterations. The final classification yielded 12 classes, of which 7 gave interpretable 3D maps. Only spatial frequencies up to 1/10 Å⁻¹ were used during refinement to avoid fitting noise to high-resolution features in maps. All seven 3D maps had Fourier shell correlation values greater than 0.8 at this frequency.

## Map analysis and model building

Segmentation of 3D maps was performed with *UCSF Chimera* (*Goddard et al., 2007*; *Pintilie et al., 2010*) and atomic structures were fit flexibly into 3D maps using *NAMD* with Molecular Dynamics Flexible Fitting (*MDFF*) (*Trabuco et al., 2008*). The $F_O$ regions from the seven different 3D maps

were aligned and averaged in real space with *UCSF Chimera* and *Situs* (*Wriggers et al., 1999*). Co-varying pairs of residues were detected in the full bovine mitochondrial ATP synthase a subunit sequence (NCBI reference YP_209210.1) with the program *EVcouplings* (*Hopf et al., 2012*) using a pseudo-likelihood maximization approach and the top 90 connections were considered in the analysis. The protein was not assumed to have trans-membrane $\alpha$-helices and the job was run as a quick launch with all other parameters at default settings. Evolutionary couplings between the a subunit and ATP synthase c subunit were detected with *GREMLIN* (*Ovchinnikov et al., 2014*) with an E-value threshold for multiple sequence alignments (MSAs) of $1 \times 10^{-10}$ and *Jackhmmer* was used to produce MSAs over 8 iterations.

To build a model of the a subunit, six straight $\alpha$-helices ($\varphi$ = -57° and $\psi$ = -47°) were built in *UCSF Chimera*. These $\alpha$-helices were fit manually in the map of the average $F_O$ region in the only orientations that satisfied constraints from evolutionary covariance analysis. For illustration, but not interpretation, loops connecting these helices were also included in the model. Randomly structured connecting loops between the $\alpha$-helices were built with *Modeller* (*Eswar et al., 2006*) within *UCSF Chimera* and fitted into the density with *MDFF* with a low density scaling factor (gscale = 0.3) over 200,000 steps (200 ps). Bond lengths and angles were then idealized with *Rosetta* (idealize_jd2 command) and the loop between residues 115 and 148 rebuilt in *Rosetta* (loopmodel command) using the quick_ccd method of remodelling (*Dimaio et al., 2009*). Each output structure included an all-atom relaxation in the density map with a score weight of 0.1. The lowest-energy model of 100 models was selected and angles were idealized and the structure energy-minimized with *UCSF Chimera*. Loops beside the one from residues 115 and 148 were too short for this process to be useful. The b subunit crystal structure was extended into the $F_O$ region of the map based on trans-membrane $\alpha$-helix prediction from MEMSAT-SVM (*Nugent and Jones, 2009*).

## Acknowledgements

We thank Richard Henderson and Voula Kanelis for a critical reading of this manuscript. A preprint of this manuscript was first deposited on bioRxiv.org (10.1101/023770) on August 11, 2015. This work was supported by operating grant MOP 81294 from the Canadian Institutes of Health Research (JLR) and Medical Research Council grant U105663150 (JW). AZ was supported by a postgraduate scholarship from the Canadian Institutes of Health Research, an award from The Hospital for Sick Children, and a U of T excellence award. DGS was supported by a postgraduate scholarship from the Natural Sciences and Engineering Research Council and an award from The Hospital for Sick Children. JLR holds the Canada Research Chair in Electron Cryomicroscopy.

## Additional information

### Funding

| Funder | Grant reference number | Author |
|---|---|---|
| Canadian Institutes of Health Research | MOP 81294 | John L Rubinstein |
| Medical Research Council | U105663150 | John E Walker |
| Howard Hughes Medical Institute | | Alexis Rohou Nikolaus Grigorieff |
| Canada Research Chairs (Chaires de recherche du Canada) | | John L Rubinstein |
| Canadian Institutes of Health Research | postgraduate scholarship | Anna Zhou |
| Natural Sciences and Engineering Research Council of Canada | postgraduate scholarship | Daniel G Schep |

The funders had no role in study design, data collection and interpretation, or the decision to submit the work for publication.

## Author contributions

AZ, AR, DGS, Acquisition of data, Analysis and interpretation of data, Drafting or revising the article; JVB, MGM, Acquisition of data, Drafting or revising the article, Contributed unpublished essential data or reagents; JEW, Conception and design, Drafting or revising the article, Contributed unpublished essential data or reagents; NG, JLR, Conception and design, Analysis and interpretation of data, Drafting or revising the article

## Author ORCIDs

Alexis Rohou, http://orcid.org/0000-0002-3343-9621

# Additional files

## Major datasets

The following datasets were generated:

| Author(s) | Year | Dataset title | Dataset URL | Database, license, and accessibility information |
|---|---|---|---|---|
| Zhou A, Rohou A, Schep DG, Bason JV, Montgomery MG, Walker JE, Grigorieff N, Rubinstein J | 2015 | Bovine mitochondrial ATP synthase state 1a | http://www.ebi.ac.uk/pdbe/entry/emdb/EMD-3164 | Publicly available at the Electron Microscopy Data Bank. |
| Zhou A, Rohou A, Schep DG, Bason JV, Montgomery MG, Walker JE, Grigorieff N, Rubinstein J | 2015 | Bovine mitochondrial ATP synthase state 1b | http://www.ebi.ac.uk/pdbe/entry/emdb/EMD-3165 | Publicly available at the Electron Microscopy Data Bank. |
| Zhou A, Rohou A, Schep DG, Bason JV, Montgomery MG, Walker JE, Grigorieff N, Rubinstein J | 2015 | Bovine mitochondrial ATP synthase state 2a | http://www.ebi.ac.uk/pdbe/entry/emdb/EMD-3166 | Publicly available at the Electron Microscopy Data Bank. |
| Zhou A, Rohou A, Schep DG, Bason JV, Montgomery MG, Walker JE, Grigorieff N, Rubinstein J | 2015 | Bovine mitochondrial ATP synthase state 2b | http://www.ebi.ac.uk/pdbe/entry/emdb/EMD-3167 | Publicly available at the Electron Microscopy Data Bank. |
| Zhou A, Rohou A, Schep DG, Bason JV, Montgomery MG, Walker JE, Grigorieff N, Rubinstein J | 2015 | Bovine mitochondrial ATP synthase state 2c | http://www.ebi.ac.uk/pdbe/entry/emdb/EMD-3168 | Publicly available at the Electron Microscopy Data Bank. |
| Zhou A, Rohou A, Schep DG, Bason JV, Montgomery MG, Walker JE, Grigorieff N, Rubinstein J | 2015 | Bovine mitochondrial ATP synthase state 3a | http://www.ebi.ac.uk/pdbe/entry/emdb/EMD-3169 | Publicly available at the Electron Microscopy Data Bank. |
| Zhou A, Rohou A, Schep DG, Bason JV, Montgomery MG, Walker JE, Grigorieff N, Rubinstein J | 2015 | Bovine mitochondrial ATP synthase state 3b | http://www.ebi.ac.uk/pdbe/entry/emdb/EMD-3170 | Publicly available at the Electron Microscopy Data Bank. |

| | | | | |
|---|---|---|---|---|
| Zhou A, Rohou A, Schep DG, Bason JV, Montgomery MG, Walker JE, Grigorieff N, Rubinstein J | 2015 | Bovine mitochondrial ATP synthase state 1a | http://www.rcsb.org/pdb/explore/explore.do?structureId=5ARA | Publicly available at the Protein Data Bank. |
| Zhou A, Rohou A, Schep DG, Bason JV, Montgomery MG, Walker JE, Grigorieff N, Rubinstein J | 2015 | Bovine mitochondrial ATP synthase state 1b | http://www.rcsb.org/pdb/explore/explore.do?structureId=5ARE | Publicly available at the Protein Data Bank. |
| Zhou A, Rohou A, Schep DG, Bason JV, Montgomery MG, Walker JE, Grigorieff N, Rubinstein J | 2015 | Bovine mitochondrial ATP synthase state 2a | http://www.rcsb.org/pdb/explore/explore.do?structureId=5ARH | Publicly available at the Protein Data Bank. |
| Zhou A, Rohou A, Schep DG, Bason JV, Montgomery MG, Walker JE, Grigorieff N, Rubinstein J | 2015 | Bovine mitochondrial ATP synthase state 2b | http://www.rcsb.org/pdb/explore/explore.do?structureId=5ARI | Publicly available at the Protein Data Bank. |
| Zhou A, Rohou A, Schep DG, Bason JV, Montgomery MG, Walker JE, Grigorieff N, Rubinstein J | 2015 | Bovine mitochondrial ATP synthase state 2c | http://www.rcsb.org/pdb/explore/explore.do?structureId=5FIJ | Publicly available at the Protein Data Bank. |
| Zhou A, Rohou A, Schep DG, Bason JV, Montgomery MG, Walker JE, Grigorieff N, Rubinstein J | 2015 | Bovine mitochondrial ATP synthase state 3a | http://www.rcsb.org/pdb/explore/explore.do?structureId=5FIK | Publicly available at the Protein Data Bank. |
| Zhou A, Rohou A, Schep DG, Bason JV, Montgomery MG, Walker JE, Grigorieff N, Rubinstein J | 2015 | Bovine mitochondrial ATP synthase state 3b | http://www.rcsb.org/pdb/explore/explore.do?structureId=5FIL | Publicly available at the Protein Data Bank. |
| Zhou A, Rohou A, Schep DG, Bason JV, Montgomery MG, Walker JE, Grigorieff N, Rubinstein J | 2015 | Bovine mitochondrial ATP synthase with averaged membrane region | http://www.ebi.ac.uk/pdbe/entry/emdb/EMD-3181 | Publicly available at the Electron Microscopy Data Bank. |

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
