## [Decision Letter]

Thank you for submitting your work entitled "Structure and conformational states of the bovine mitochondrial ATP synthase by cryo-EM" for peer review at *eLife*. Your submission has been favorably evaluated by John Kuriyan (Senior Editor), and two reviewers, Stephen Harrison (also Reviewing Editor) and Yifan Chang.

The reviewers have discussed the reviews with one another and the Reviewing Editor has drafted this decision to help you prepare a revised submission.

This very thoughtful cryoEM analysis of the bovine mitochondrial ATP synthase at subnanometer resolution is impressive for its thoughtful approach and for the straightforwardness of its conclusions. The authors took full advantages of single particle cryo-EM for dealing samples with significant conformational heterogeneity, and identified seven different conformational states of the ATP synthase enzyme. The resolutions of these cryo-EM reconstructions are in the sub-nanometer range, sufficient to resolve most, if not all, secondary structural features. The flexible fitting is well described, and the restriction of interpretation to well-resolved helices is laudable. The requested revisions below are purely at the level of presentation and should be straightforward to implement.

1) The authors identified a total of seven states, 3 sub-states for state 1, 2 for state 2 and 3 for state 3. It is very difficult to visualize the significant differences between these sub-states from the figures presented. Thus, it is difficult to appreciate the subtle conformational changes and the significances represented by these sub-states. The manuscript does not include any quantitative descriptions of these conformational changes. Showing two overlapping maps with different colors might make the differences more obvious, and a quantitative description of the conformational differences might likewise be useful. The authors should choose some way of making the differences clear.

2) A related question: how are these sub-states connected and do they have any functional significance, or are they simply evidence of the distribution of states in a "flat" well around the equilibrium position? It is easier to identify the sequence of conformational changes between major states, 1, 2 and 3.

3) At the end of the second paragraph of the Results, a feature with "unusually low density" is interpreted as lipids. This is not very convincing. Lipid density confined in such a small region may not be that weak. Is there any additional evidence for this interpretation (perhaps from the molecular structure achieved in the end by fitting known components)?

4) The authors averaged seven maps to enhance the SNR of the membrane region. This averaging certainly enhances SNR of certain parts of the molecule, where the conformation remains unchanged, but it of course weakens the other parts. This point, obvious to some readers, should be stated clearly, as it also relates to question above: why it is critical to separate these small conformational differences (other than to get maps that can yield good local averages? Indeed, have the authors considered local averaging of other regions, using suitable rigid-body transformations based on the fitting of models? (This might go beyond the present paper, but the comment is worth considering.)

5) The authors should include raw data as supplements to one or more of the figures: micrographs, power spectra of the micrographs, Euler angle distributions, and 2D class averages, all of which are needed for a reader to judge the quality of cryo-EM results.

6) The authors should provide some additional description at various points to help readers unfamiliar with the details of the ATP synthase structure. For example, in the second paragraph of the Results, the paper discusses "a distinct bend". It is hard to figure out which part of the text is about previous structures and which part is about the current structure. For readers not familiar with this complicated complex, it is also not clear which part of the illustrated structure is F0 and which part is F1. It will be useful to mark this clear in a figure, either as part of Figure 1 or as a supplementary figure. As a further example of assisting a reader not familiar with mitochondrial protein structures, one reviewer asked: in the subsection "A novel feature in the F0 region", what does "inter-membrane" mean?

---

## [Author Response]

1) The authors identified a total of seven states, 3 sub-states for state 1, 2 for state 2 and 3 for state 3. It is very difficult to visualize the significant differences between these sub-states from the figures presented. Thus, it is difficult to appreciate the subtle conformational changes and the significances represented by these sub-states. The manuscript does not include any quantitative descriptions of these conformational changes. Showing two overlapping maps with different colors might make the differences more obvious, and a quantitative description of the conformational differences might likewise be useful. The authors should choose some way of making the differences clear.

We have now included a supplement to Figure 3 that illustrates the conformation differences between sub-states in two different ways. We also include the following quantification statement in the text:

“The largest change between the sub-states of each state can be approximated by rotations of the α_3_β_3_ hexamer relative to the rest of the complex by angles ranging from 10 to 16°. A comparison of the maps for the different sub-states and the axes of these rotations are shown in Figure 3—figure supplement 1. The resulting conformational changes can be seen most clearly in Video 4 and Video 5”

The caption for Figure 3—figure supplement 1 provides further quantification of these rotations.

*2) A related question: how are these sub-states connected and do they have any functional significance, or are they simply evidence of the distribution of states in a "flat" well around the equilibrium position? It is easier to identify the sequence of conformational changes between major states, 1, 2 and 3.*

We have added the following text to the Results section:

“The different sub-states do not appear to have a specific sequence or represent specific intermediates in the rotation sequence. Instead, the differences in conformation between sub-states when taken together illustrate the flexibility of the enzyme, a property already linked to its rapid rate of enzymatic activity (_8_-ring with respect to the subunit, as discussed below.”

The Discussion section describes the rotational variation:

“This lack of a rigid interaction between the c_8_-ring and a subunit is consistent with the Brownian ratchet model of proton translocation (23). In the Brownian ratchet model […]”

We have also added the following to the Discussion section:

“Therefore, with this model, the different sub-states would correspond to energetically equivalent or nearly-equivalent conformations that occur due to Brownian motion.”

*3) At the end of the second paragraph of the Results, a feature with "unusually low density" is interpreted as lipids. This is not very convincing. Lipid density confined in such a small region may not be that weak. Is there any additional evidence for this interpretation (perhaps from the molecular structure achieved in the end by fitting known components)?*

We are still unsure of what is causing the appearance of this part of the map. We can find no insight from the structure or by fitting known components. We cannot identify any literature that lends support to the lipid hypothesis or any other hypothesis. We have expressed this uncertainty by modifying the section to read:

“The content of this low-density region is uncertain. […] Instead, the low density could be due to bound material with low density, possibly lipid, that remains after purification of the enzyme.”

*4) The authors averaged seven maps to enhance the SNR of the membrane region. This averaging certainly enhances SNR of certain parts of the molecule, where the conformation remains unchanged, but it of course weakens the other parts. This point, obvious to some readers, should be stated clearly, as it also relates to question above: why it is critical to separate these small conformational differences (other than to get maps that can yield good local averages? Indeed, have the authors considered local averaging of other regions, using suitable rigid-body transformations based on the fitting of models? (This might go beyond the present paper, but the comment is worth considering.)*

We agree completely with the comment and have re-written this section of the manuscript:

“In order to improve the signal-to-noise ratio for the F_O_ region of the complex […] suggesting that its position differs between maps (Figure 1, pink arrows).”

*5) The authors should include raw data as supplements to one or more of the figures: micrographs, power spectra of the micrographs, Euler angle distributions, and 2D class averages, all of which are needed for a reader to judge the quality of cryo-EM results.*

We now include Figure 1—figure supplement 1 that shows a portion of a raw micrograph, power spectrum, 2D class averages, particle trajectories from the micrograph from the *alignparts_lmbfgs* optimization, and the distribution of Euler angles from particles that went into the 3D maps.

6) The authors should provide some additional description at various points to help readers unfamiliar with the details of the ATP synthase structure. For example, in the second paragraph of the Results, the paper discusses "a distinct bend". It is hard to figure out which part of the text is about previous structures and which part is about the current structure. For readers not familiar with this complicated complex, it is also not clear which part of the illustrated structure is F0 and which part is F1. It will be useful to mark this clear in a figure, either as part of Figure 1 or as a supplementary figure. As a further example of assisting a reader not familiar with mitochondrial protein structures, one reviewer asked: in the section "A novel feature in the F0 region", what does "inter-membrane" mean?

We have re-written the description of the bend in the F_O_ region to more clearly distinguish past and present results:

“There is a distinct bend in the F_O_ region of the complex between the portion that is proximal to the c_8_-ring […] was also observed recently by electron tomography of membrane-reconstituted 2D crystals of the bovine enzyme (22).

We have modified Figure 1 to indicate the F_1_ region and the F_O_ region.

We have also modified Figure 1—figure supplement 2 (formerly supplement 1) to indicate the bend in the F_O_ region.

To clarify the term “inter-membrane space” we have added a sentence to the Introduction:

“The ATP synthase is found in the inner membranes of mitochondria, with the F_1_ region in the mitochondrial matrix and the F_O_ region accessible from the inter-membrane space between the mitochondrial outer and inner membranes.”